

**Diversity and mineral substrate preference in endolithic microbial communities**
**from marine intertidal outcrops (Isla de Mona, Puerto Rico).**
Estelle Couradeau[1,2], Daniel Roush[1], Brandon Scott Guida[1], Ferran Garcia-Pichel[1]
[1]School of Life Sciences, Arizona State University, 85282 Tempe, Arizona, USA
[2]Laboratoire Biogéosciences, UMR6282, Université de Bourgogne, 21000 Dijon, France
**Corresponding author:** Ferran Garcia-Pichel ferran@asu.edu
**Running title:** endolithic cyanobacteria substrate preference
**Abstract**
Endolithic microbial communities are prominent features of intertidal marine habitats, where they
colonize a variety of substrates, contributing to their erosion. Almost two centuries worth of naturalistic
studies focused on a few true-boring (euendolithic) phototrophs, but substrate preference has received
little attention. The Isla de Mona (Puerto Rico) intertidal zone offers a unique setting to investigate
substrate specificity of endolithic communities since various phosphate rock, limestone, and dolostone
outcrops occur there. High-throughput 16S rDNA genetic sampling, enhanced by targeted cultivation,
revealed that, while euendolithic cyanobacteria were dominant, the communities were invariably of
high diversity, well beyond that reported in traditional studies, and implying an unexpected metabolic
complexity, potentially contributed by secondary colonizers. While the overall community composition
did not show differences traceable to the nature of the mineral substrate, we detected specialization



among particular euendolithic cyanobacterial clades towards the type of substrate they excavate, but
only at the OTU phylogenetic level, implying that close relatives have specialized recurrently into
particular substrates. The cationic mineral component was determinant in this preference, calling for
the existence in nature of alternatives to the boring mechanism described in culture that is based
exclusively on transcellular calcium transport.





**Introduction**
In shallow and intertidal marine habitats, endolithic microbes colonize a variety of carbonaceous and
phosphatic substrates, such as bone, shell, coraline carbonate, ooliths, as well as limestones, dolostone
and phosphorite outcrops (Campbell, 1983). Some of these microbes take advantage of the natural
pores or crevices in the solids, but some have the ability to actively bore their way into the substrate.
Such microborers, also known as euendoliths (Golubic et al., 1981), build communities that can cover
as much as 50% of the exposed solid surface (Golubic et al., 2000) with full colonization times of
virgin substrate on the order of months (Gektidis, 1999; Grange et al., 2015). Several long-term
geological phenomena are driven by microborers, from the erosive morphogenesis of coastal
limestones (Purdy and Kornicker, 1958; Schneider, 1983; Torunski, 1979; Trudgill, 1987) and the
destruction of coral reefs and other biological carbonates (Le Campion-Alsumard et al., 1995;
Ghirardelli, 2002) to the cementation of loosely bound carbonate grains in coastal stromatolites
(MacIntyre et al., 2000; Reid et al., 2000). Additionally, phototrophic euendoliths can cause significant
damage and shell weakening to bivalve populations (Kaehler and McQuaid, 1999). Long-term rates of
microborer-driven carbonate dissolution, the "bioerosion" process, range between 20 and 930 g $CaCO_3$
$m^{-2}$ $d^{-1}$, are of clear geologic significance (Grange et al., 2015; Peyrot-Clausade et al., 1995; Tudhope
and Risk, 1985; Vogel et al., 2000), and may increase under future scenarios of increased atmospheric
$CO_2$ and ocean acidification (Tribollet et al., 2009).

There exists a very large body of descriptive literature spanning 18 decades, largely based on
microscopic observations, documenting the biodiversity of microborers, with contributions in the
microbiological, ecological, sedimentological and paleontological fields (Acton, 1916; Al-Thukair et
al., 1994; Bachmann, 1915; Batters, 1892; Bonar, 1942; Bornet and Flahault, 1888; Budd and Perkins,
1980; Le Campion-Alsumard et al., 1995; Chodat, 1898; Duerden, 1902; Duncan, 1876; Ercegovic,



1925, 1927, 1930, Frémy, 1936, 1941; Ghirardelli, 2002; Golubic, 1969; Kölliker, 1859; Lehmann,
1903; May and Perkins, 1979; Nadson, 1927; Pantazidou et al., 2006; Perkins and Tsentas, 1976;
Wisshak et al., 2011). Euendoliths have been reported among eukaryotes (fungi, green and red algae)
and prokaryotes (cyanobacteria). The most common genera of phototrophic eukaryotic euendoliths are
*Ostreobium* and *Phaeophila* in the green algae, as well as the red algal genus *Porphyra* (in its
filamentous diploid generation, known also as *Conchocelis* stage). In the cyanobacteria, the
pseudofilamentous genera *Hyella* and *Solentia* are quite common (Al-Thukair, 2011; Al-Thukair et al.,
1994; Al-Thukair and Golubic, 1991; Brito et al., 2012; Campion-Alsumard et al., 1996; Foster et al.,
2009; Golubic et al., 1996) , as are some forms in the simple filamentous genus *Plectonema* (Chacón et
al., 2006; Pantazidou et al., 2006; Tribollet and Payri, 2001; Vogel et al., 2000). Morphologically
complex cyanobacteria such as *Mastigocoleus testarum* (Golubic and Campion-Alsumard, 1973;
Nadson, 1932; Ramírez-Reinat and Garcia-Pichel, 2012a) complete the list of common euendoliths.
Less common genera of euendolithic cyanobacteria include: *Cyanosaccus* (Pantazidou et al., 2006),
*Kyrtuthrix* (Golubic and Campion-Alsumard, 1973) and *Matteia* (Friedmann et al., 1993). These genera
were all assigned based upon morphological criteria and could represent morphological variations of
the same types (Le Campion-Alsumard and Golubic, 1985), highlighting the need to re-assess the
diversity of euendolithic cyanobacteria using a combination of characters including genetic markers, a
task yet to be undertaken with any breadth.

Modern genomic methods for community fingerprinting have, more recently, been applied to provide
an alternative, comprehensive description of endolithic communities. Some studies, focused on
phototrophs from marine carbonates, revealed that, while some biodiversity had been missed by
deploying merely morphological studies, there was also congruency between DNA-based surveys, and
the traditional literature (Chacón et al., 2006; Ramírez-Reinat and Garcia-Pichel, 2012b). DNA-based
studies brought to our attention that the endolithic habitat at large can harbor complex communities of





microbes, not just composed of euendoliths, particularly when the substrate rocks are naturally porous,
or when they have been rendered porous by the action of euendoliths themselves. Horath and Bachofen
2006, for example, investigating terrestrial endolithic communities in dolomite outcrops in the Alps,
found a large diversity of presumably chemotrophic bacteria and archaea, in addition to expected green
algae and cyanobacteria. Similar conclusions could be drawn from the work of de la Torre et al. (De la
Torre et al., 2003) on Antarctic sandstone cryptoendoliths, those of Walker and colleagues (Walker et
al., 2005; Walker and Pace, 2007) on terrestrial limestones, sandstones and granites or the recent
contribution of (Crits-Christoph et al., 2016) who used a metagenomic approach to investigate the
chasmoendolithic communities of the hyper-arid Atacama desert. However, no studies are yet available
on the globally significant intertidal endolithic communities that have used the power of high-
throughput sequencing techniques.

Tribollet (2008) provided an account of the dynamic changes in microborer community composition
taking place after coral death, which obviously constitute a true succession in the ecological sense, with
pioneer euendoliths (such as *Mastigocoleus testarum*) and secondary colonizers such as *Ostreobium*
*quekettii* and *Plectonema terebrans,* as well as fungi (Grange et al., 2015; Tribollet, 2008). During
laboratory studies with the cultivated strain of *Mastigocoleus testarum* strain BC008, used as a model
to understand the physiology of cyanobacterial boring (Garcia-Pichel et al., 2010; Guida and Garcia-
Pichel, 2016; Ramírez-Reinat and Garcia-Pichel, 2012b), we could show that, among the carbonates,
this strain excavated fastest into various types of calcite and aragonite minerals ($CaCO_3$). It could bore
slowly into strontianite ($SrCO_3$), but was unable to penetrate into magnesite ($MgCO_3$), dolomite
($CaMgCO_3$), witherite ($BaCO_3$), rhodochrosite ($MnCO_3$), siderite ($FeCO_3$) or ankerite
($CaFe(CO_3)_2$)(Ramírez-Reinat and Garcia-Pichel, 2012a). However, literature reports do exist detailing
microborings in modern and fossil dolomitic substrates (see e.g. (Campbell, 1983; Golubic and Lee,
1999). Similar arguments can be made for phosphates: *M. testarum* strain BC008 did not bore into





calcophosphatic substrates, including hydroxyapatite, vivianite or dentine; yet, the literature is replete
with reports of cyanobacterial microborings on biotic and abiotic phosphatic rocks (Soudry and Nathan,
2000; Underwood et al., 1999; Zhang and Pratt, 2008)). The expression of such a mineral substrate
preference among the pioneer euendolithic cyanobacteria could principally drive the whole community
towards a different successional sequence with distinct mature community assemblages and metabolic
potentialities. We wanted to ask the question if evolutionary specialization has resulted in a highly
adapted endolithic flora for each type of mineral substrate, and if there exist specialized apatite-borers,
dolomite-borers, or carbonate-borers in nature. Surprisingly, this aspect of endolithic microbiology had
not been directly addressed yet.

In order to answer these questions, we investigated in depth the marine endolithic communities of Isla
de Mona (PR), a small, uninhabited Caribbean island offering a variety of coastal cliffs composed of
dolomite and limestone, as well as raised aragonitic and phosphatic reefs, with the dual purpose to (i)
describe the microbial diversity of intertidal endolithic community at high resolution and (ii) to test the
effects of substrate composition on community structure in a single geographic location with common
bathymetry (the intertidal notch), controlling for other known major determinants of community
composition.

**Materials and Methods**

*Sampling site and procedure*

Samples were obtained from Isla Mona (18.0867° N, 67.8894° W), a small (11 km by 7 km) carbonate
island 66 km W of Puerto Rico. Isla Mona is a protected habitat and all necessary permits were ac-
quired from the Departamento de Recursos Naturales y Ambientales prior to arrival. The present study



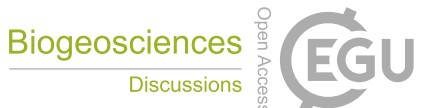

did not involve endangered or protected species. Endolithic communities were obtained by sampling
different locations from nine separate island localities. Rock samples containing endolithic biomass,
verified using a digital field microscope, were chipped off from large boulders and rock walls using a
standard geological hammer. Material was predominantly collected within the boring notch of the inter-
tidal zone. Bathymetric samples were collected via SCUBA diving at sample site K at depths of 3.5,
4.6, 7, and 9.1 meters. Three replicates were taken per sample which consisted of sterile 50 mL falcon
tubes filled with material, one replicate was air dried for mineralogical analysis, one was kept viable in
seawater for strain isolation and another was preserved *in situ* in 70% ethanol for DNA extrac-
tion. Samples were shipped at room temperature, and, upon arrival in the lab, the preserved samples
were immediately stored at -20$^{\circ}$C until extractions were performed. Aliquots of local seawater were
filtered through 0.22 µm syringe filters into sterile 50 mL falcon tubes for physico-chemical analysis.

*Bulk powder X ray diffraction and elementary analyses*

A fragment of each sample was ground down to powder in 100% ethanol. XRD patterns were collected
using Panalytical X'Pert Pro diffractometer mounted in the Debye-Scherrer configuration with a CuKα
monochromatic X-Ray source. Data were recorded in continuous scan mode within a 10–90° 2θ range.
X'Pert High Score plus software was used to identify mineral phases and retrieved their relative
concentration using the automatic Rietveld refinement method implemented in the software under
default parameters. The elementary composition of the rocks and water sample analyses were
performed by the Goldwater Center at Arizona State University using a Inductively Coupled Plasma
Optical Emission Spectrometer (ICP-OES) - Thermo iCAP6300.

*Total genomic DNA purification*



The surface of the ethanol fixed samples was brushed vigorously with a sterile toothbrush and sterile
MilliQ water to remove epilithic material. A chip of 8 cm$^3$ was further grounded in a sterile mortar as
recommended by (Wade and Garcia-Pichel, 2003). 0.5 g of the obtained coarse powder was then
transferred into the bead tube of the MoBio PowerPlant Pro kit (Mo Bio Laboratories, Inc., Carlsbad,
CA, USA). The first lysis step of the kit was modified as follow bead tubes were homogenized
horizontally at 2,200 rev/min for 10 minutes and 7 freeze-thaw cycles were applied (Wade and Garcia-
Pichel, 2003). The next steps of the extraction were conducted following the MoBio PowerPlant Pro kit
following manufacturer's guidelines.

*16s rRNA gene library preparation and sequencing*

The 16S rRNA gene V3 - V4 variable region was targeted using PCR primers 341F
(CCTACGGGNGGCWGCAG ) and 806R (GGACTACVSGGGTATCTAAT) with a barcoded forward
primer. The PCR amplification was performed using the HotStartTaq Plus Master Mix Kit (Qiagen,
USA) under the following conditions: 94°C for 3 minutes, followed by 28 cycles of 94°C for 30
seconds, 53°C for 40 seconds and 72°C for 1 minute, followed by a final 5min elongation step at 72°C.
PCR product were further purified and pooled into a single DNA library using the Illumina TruSeq
DNA library preparation protocol. This library was further sequenced on a MiSeq following the
manufacturer's guidelines. The library preparation, sequencing paired ends assembly and first quality
trimming (with phred score of Q25 cutoff) was performed by MR DNA (www.mrdnalab.com,
Shallowater, TX, USA).

*OTU table building and analysis*

Sequences were further processed using the Qiime version 1.9 (Caporaso et al., 2010). The sequences



were first run through the *split_libraries.py* script under the default parameter that includes barcodes
removal, quality filtering (sequences of less than 200bp or with homopolymer runs exceeding 6bp were
removed) and split of the dataset per sample. The output file was further process through the
*pick_open_reference_otus.py* script using the default parameters except for the taxonomic assignment
that was done by the RDP classifier (see parameter file in supplementary information for more details).
This step clustered the sequences at a similarity threshold of 97% (Edgar, 2010) to build Operational
Taxonomic Units (OTUs), assign their taxonomy and reported their specific abundance in each sample
into an OTU table. Because in this case we were not interested into the rare biosphere but focused on
the most abundant OTUs and how they vary, we filtered the OTU table to remove the rare OTUs. The
OTUs retained were those that occurred in at least 5 samples among the 34 analyzed, or that represent
more than 0.1% of the total sequences found in a particular sample. By doing this, we eventually
analyzed 90% of all the single sequences but only 11% of the initial OTUs.  The Qiime script
*summarize_taxonomy_through_plots.py* was run on the final OTU table for all the prokaryotes and for
the Cyanobacteria only (filtering out the chloroplasts) in order to build the summarized microbial
community composition bar graphs displayed on the figure 2. One representative sequence per OTU
was deposited to genebank under the accession numbers KT972744-KT981874.

*Accession numbers*
One representative sequence per OTU was deposited to genebank under the accession numbers
KT972744-KT981874. The 16S rDNA sequences of the new euendolithic strains described in this
article received the following accession numbers: *Ca.* Pleuronema perforans IdMA4 [KX388631], *Ca.*
Mastigocoleus perforans IdM [KX388632], *Ca.* Pleuronema testarum RPB [KX388633].

*Meta-analysis of microbial communities*



Raw sequences from datasets ID 662/678/809/627/713/925 were retrieved from the Qiita repository
along with their mapping table. All these studies used comparable sequencing depth, technology and
targeted the same region of the 16 rRNA gene compared to the present study. Two samples from
Alchichica cyanobacteria dominated microbialites communities (Couradeau et al., 2011) were
processed in parallel to the Isla de Mona samples (same extraction methodology, sequenced in the same
MiSeq run), they were included in this analysis as well. The sequences were all aggregated into a
masterfile that was processed in Qiime version 1.9 (Caporaso et al., 2010). The same exact procedure
than the one described above was used to pick OTUs. Again we retained the OTUs that occurred at
least in 5 samples. We ran the *jackknifed_beta_diversity.py* pipeline using the Bray Curtis metrics
under default parameters. The obtained distances were used to cluster samples under a UPGMA
hierarchical clustering method and 5000 sequences were included in each jackknifed subset in order to
generate nodes support.

*Differential abundance of OTUs analyses*

To determine if some OTUs were more associated to certain type of substrates we run the
*differential_abundance.py* of the Qiime 1.9 package (Caporaso et al., 2010) using the DESeq2 method
(Love et al., 2014), under a negative binomial generalized linear model. This method was initially
developed to assess the differential gene expression from RNA seq data but can be applied to any count
matrix data such as OTU tables (Love et al., 2014). It was recently implemented for the treatment of
16S rDNA OTU table and as been widely used since (e.g. (Debenport et al., 2015; Pitombo et al.,
2015)) because it (i) is a sensitive and precise method, (ii) controls the false positive rate (Love et al.,
2014) and (iii) it uses all the power of the dataset without the need to rarefy the OTU table (McMurdie
and Holmes, 2014). After checking the good agreement between the fit line and the shrinked data on
the dispersion plot, a Wald test was applied to each OTU to reject the null hypothesis ($p < 0.05$) being





that the logarithmic fold change between treatments (i.e. in our case type of mineral substrate) for a
given OTU is null.

*Phylogeny reconstruction*

In order to determine which of the cyanobacterial OTUs of the dataset were possible euendolithic
organisms, we built a phylogeny to assess their proximity to proven boring cultured strains. The
maximum-likelihood phylogenetic reconstruction was performed using TREEFINDER (Jobb et al.,
2004) under a general time reversible (GTR) and a four-category discrete approximation of a Γ
distribution. Bootstrap values were inferred from 1000 replicates. The sequence dataset used for the
reconstruction was first aligned with MAFFT (Katoh et al., 2005) and then manually checked and
trimmed using the MUST package (Philippe, 1993).

**Results & Discussion**

*Geological setting of Isla de Mona outcrops.*

The island is an 11 by 7 km emerged platform of Miocene Isla de Mona Dolomite (up to 80 m thick)
topped by a thinner (up to 40 m) layer of Miocene Lirio limestone (Briggs and Seiders, 1972; Frank et
al., 1998). It is partially surrounded in its Southern and Southwestern shores by a Pleistocene raised
reef flat, mostly composed of biogenic carbonates (Fig. 1). The island also harbors secondary
phosphorite deposits formed by the diagenetic alteration of guano, most typically associated with an
extensive system of karstic caves at the interface of limestone and dolostone (Briggs, 1959). Isla de
Mona was never continuously inhabited, mostly used as a guard post over the Mona Passage
throughout the 20[th] century, and declared a Nature Preserve in 1993 (National Parks Register, USA).



The coastal area has been protected from disturbance ever since. We took advantage of this unique and
pristine geological setting to sample dolostones, limestones and phosphorites exposed to similar
environmental conditions. We analyzed a set of 34 samples consisting of pieces of exposed rock, in
most cases taken directly at the intertidal notch. Location of sampling sites are in the simplified
geological map in Figure 1a. The mineralogical composition of each sample (Fig. 2), determined using
bulk powder X-Ray diffraction, confirmed the presence of apatite ($Ca_5(PO_4)_3(OH,Cl,F)$), dolomite
($CaMg(CO_3)_2$), calcite ($CaCO_3$) and aragonite($CaCO_3$) in various proportions depending of the
sampling site (Fig. 2a).

*The endolithic microbial communities*

We studied the endolithic community composition by analyzing the 16S rDNA diversity present in total
genomic DNA extracted from the rock after aggressively brushing away epilithic growth from the
external sample surface. The 16S rDNA sequences were obtained after specific PCR amplification and
Illumina-based high-throughput sequencing, with one library per sample (Table S2). We clustered
sequences into OTUs (Operational Taxonomic Units) based on a 97% similarity criterion, and further
filtered the dataset to remove the rare OTUs, focusing our study on OTUs that occurred in at least five
separate samples, or those that made up more than 0.1% of all sequences in any one sample. Bacterial
OTU richness in these samples was 4058 ±1252, as given by the chao1 metric (Figure 2c). Thus,
comparatively our endolithic communities are of rather low diversity, an order of magnitude lower than
current estimates assigned to bulk soil bacterial communities (Roesch et al., 2007), but similar to other
microbial communities such as biological soil crusts (Couradeau et al., 2016), microbial mats
(Hoffmann et al., 2015) or stromatolites (Mobberley et al., 2011), that are dominated by cyanobacterial
primary producers. This suggests that endolithic habitat nurtured by the presence of cyanobacterial
primary producers can support the development of a high diversity of microorganisms even if this type



of habitat is expected to be nutrient limited due to its low connectivity with sea water (Cockell and
Herrera, 2008). Taxonomic assignment of the OTUs on the basis of the Greengene database (McDonald
et al., 2012), allowed us to reconstruct the endolithic prokaryotic communities from Isla de Mona at
various level of taxonomic resolution. At the phylum level (Figure 2b), the analysis revealed complex
microbial communities with numerically very significant populations of bacteria other than
Cyanobacteria: *Proteobacteria, Chloroflexi, Actinobacteria* and *Bacteroidetes*. In fact, the contribution
of cyanobacteria to the total sequence richness was only $12 \pm 3\%$. These communities clearly host not
only a large number of bacterial types, but also a wide diversity of phylogenetic and metabolic
potential beyond oxygenic photosynthesis. Clearly, mature endolithic cyanobacterial communities are
much more complex than the overwhelming majority of the traditional literature would suggest (for
example, the exhaustive descriptive literature review in the introduction does not report beyond
cyanobacteria and eukaryotic algae). While it is proven by the use of model organisms in culture that
cyanobacteria alone are able to initiate excavation on virgin substrate (Ramírez-Reinat and Garcia-
Pichel, 2012a), it is interesting to entertain that in such complex communities, other metabolic
activities, particularly those that result in pH changes might play a significant role on the determination
of the local saturation index of the carbonate mineral (Baumgartner et al., 2006; Dupraz et al., 2009;
Dupraz and Visscher, 2005), and in this way influence the overall mineral excavation yield or rates. At
this level of taxonomic resolution, we did not detect any significant association of substrate mineralogy
and community composition (as judged by non significant Spearman's ρ when comparing each
phylum's relative abundance to mineralogical composition, not shown).

Because endolithic communities have not received much attention, we integrated our dataset into a
meta-analysis of various cognate microbial communities, for which technically comparable datasets
were publicly available (http://qiita.microbio.me.). To do so, we aggregated all the sequences from the
selected Qiita datasets into a single file that was used to pick and cluster 16S rDNA OTUs anew, and





conducted similarity analyses. The meta-community analysis revealed that endolithic communities
clustered together, and apart from other types of phototrophic microbial communities in terms of
composition (beta-diversity). The fact that they clustered together indicates that their microbial
assemblages are recognizable and distinct beyond just their belonging to the marine habitat itself, in a
microbiological and presumably adaptive way. A cautionary alternative reading, however, could be that
this pattern represents a biogeographical island effect, in that all of our samples come from a relatively
small geographical area. This alternative explanation is unlikely given the cosmopolitan nature of
marine cyanobacteria (Garcia-Pichel et al., 1996; Lodders et al., 2005) Interestingly, our endolithic
community samples could be separated into 2 self-similar clades (A and B Figure 3) but so far we
cannot ascertain a factor that would drive the observed separation beyond the fact that it is not substrate
type. While it would be of interest to compare our communities to other endolithic communities, such
as those studied by (Chacón et al., 2006; Crits-Christoph et al., 2016; Horath and Bachofen, 2009; De
la Torre et al., 2003) this is not technically possible, given that all of those studies used alternative
methods for community analyses (Clone libraries, DGGE, metagenomes) that do not allow direct
comparisons.

*A diverse cyanobacterial community dominated by likely euendoliths*

Because they comprise the pioneer microborers and primary producers within many endolithic
communities, cyanobacteria are of particular interest in this study. We therefore analyzed cyanobacteria
at a higher resolution. The cyanobacterial community appeared quite diverse with a specific chao1
richness of 484 ±184, certainly much more genetic diversity among this group than could be surmised
from the wealth of microscopically based accounts in the botanical literature (Chazottes et al., 1995;
Pantazidou et al., 2006; Sartoretto, 1998; Tribollet et al., 2006). In these studies typically one finds
reports of anywhere from 1 to 5 morphotypes. Even accounting for the fact that morphotypes typically



underestimate genetic diversity by a significant fraction (Nübel et al., 1999) this is a very large
underestimation of oxygenic phototroph diversity. Phylotypes assignable to the orders
*Pseudanabaenales, Chroococcales, Nostocales* and *Stigonematales* were most common and
widespread. Again no pattern linking mineralogy to microbial community composition arose at this
taxonomic level, as judged by the non-significant Spearman's $\rho$ when comparing the relative
abundance of each cyanobacterial to mineralogical composition (not shown). A combination of
literature search and additional efforts of cultivation and genetic characterization of isolates, allowed us
to attempt the assignment of a true-boring (euendolithic) role to some of our cyanobacterial OTUs
(Table 1 and Figures S2-S3). Interestingly, out of the five most abundant OTUs in our combined
dataset, four (NR_OTU741, OTU 842393, NR_OTU193 and OTU 351529) could be deemed as likely
euendoliths, given their close phylogenetic affiliation to cultivated isolates proven in the laboratory to
be able to bore. The fifth most abundant OTU (OTU 186537) fell between *Mastigocoleus testarum*
BC008 (a proven euendolith) and *Rivularia atra* (not described as boring in the literature), so its
capacities remain unclear. Notably, the most abundant OTU, NR_OTU741 in our set is virtually
indistinguishable from one of our isolates obtained from the same samples, the boring strain *Ca.*
Pleuronema perforans IdMA4 (similarity > 99%), which is not only the most abundant cyanobacterial
OTU but also the second most abundant bacterial OTU overall in our dataset. These results suggest that
euendoliths compose a major fraction of the community, one that does not only represent an initial set
of pioneers, but one that maintains relevance even after bioerosive degradation and reworking of the
mineral substrates allow the colonization of newly made pore spaces by non-boring endoliths.

On analyzing the diversity of the possible euendoliths detected in this dataset, we realized that while
many of the most common known genera of cyanobacterial microborers are represented and abundant,
the thin, filamentous *Plectonema terebrans* is not. This was surprising because *Plectonema terebrans*
has always been described as an important member of the euendolithic community who can account for





up to 80% of the total of microborer biomass (Tribollet, 2008) and is found associated to *Mastigocoleus*
*testarum*. This apparent paradox is likely not due to the absence of the organism, but to failure to
properly identify it molecularly, due to the lack of reference sequences in the databases. Indeed
morphotypes resembling *Plectonema terebrans* was visually recognized, but not detected molecularly
in the extensive study of euendolithic cyanobacteria from various locations by (Ramírez-Reinat and
Garcia-Pichel, 2012b). In the present dataset *Plectonema* could have been assigned to another member
of the Oscillatoriales, such as *Phormidium* or *Halomicronema,* which represent 10 and 4.6% ,
respectively, of the cyanobacterial sequences. A *bona fide* isolate proven to bore in the lab will be
needed before we can advance regarding the presence and abundance of simple filamentous
euendolithic cyanobacteria anywhere. Among the cyanobacterial taxa detected, the following have
never been reported to be true borers: Gloeobacterales, Nostocaceae, Acaryochlorales,
Cyanobacteriaceae, Spirulinaceae, Pseudanabaenales. In all, these cyanobacteria contribute at least to
some 43 ±20 % indicating that a significant proportion of the community is likely made up of
adventitious endoliths. A study of the temporal dynamics of colonization could help understand the true
role of each taxon.

*Substrate preference among cyanobacteria*

We knew from the experimental study of the model euendolith *Mastigocoleus testarum* strain BC008,
that this particular organism exhibits a clear boring substrate preference. It bores into Ca-carbonates
(like aragonite and calcite) and to a lesser extent Sr-carbonate (strontianite), but not into CaMg-
carbonate like dolomite (Ramírez-Reinat and Garcia-Pichel, 2012a). This strain remains the single case
where the boring preference has been directly tested, but it is unknown if this preferential behavior is
representative of euendoliths at large. Only a few studies examined endolithic communities colonizing
dolostone, (Jones, 1989) provided the first comparison of endolithic communities from dolostones and



limestones from Grand Cayman Ironshore. He observed that dolostones were less colonized by
endoliths than limestones and concluded that the bioerosion of limestones was faster due to the more
abundant endolithic flora while the erosion pattern of the dolostone was slower and allowed the
development of more epiliths. When looking at the endolithic microbial diversity of terrestrial
dolostones (Horath et al., 2006) found the same cyanobacterial genera than the ones typically described
on freshwater limestones substrates (Norris and Castenholz, 2006) while (Sigler et al., 2003) concluded
that the endolithic dolostone phototrophic community resembled other desiccation-tolerant endolithic
communities. The question of whether there really exists a specialized community associated to
dolostone *vs.* limestone remained clearly open.

Our own data showed no specificity for substrate at family level, highlighting the need to analyze this
at a phylogenetically deeper resolution. To do so, we analyzed how cyanobacterial OTUs where
differentially represented in sample subsets from contrasted mineralogical substrates using the DESeq2
method (Love et al., 2014). This method was developed to analyze RNA-seq datasets but can be used
on any count matrix such as an OTU table. This statistical framework is sensitive and precise and does
not involve rarefying the dataset to an even sampling depth, so that the entire statistical power of the
data is accounted for (McMurdie and Holmes, 2014). We used it to determine whether any given OTU
is significantly differentially represented in a particular subset of samples sharing a common
mineralogical substrate compared to another set. In comparing OTU detected in samples were
mineralogically dominated by Ca-carbonates (calcite or aragonite, n=13) with those that were dolomitic
in nature ( CaMg-carbonate, n=14), we found 31 OTUs to be were significantly enriched in Ca-
carbonate substrates (p<0.05; corresponding to $\log_2$ fold difference $> |2.83|$), while 22 preferred
dolomite with p<0.05, out of 1039 cyanobacterial OTUs considered. It becomes clear that substrate
preferences are indeed found when one looks at fine taxonomic resolution, and that some likely
euendoliths show such preference: *Mastigocoleus testarum* close relative NR_OTU193 prefers the Ca-



carbonate pole ($\log_2$ fold difference = |3.4|) while another possible euendolith NR_OTU741 belonging
to the *Pleurocapsales* clearly prefers dolomite ($\log_2$ fold difference = |1.7|). It is also clear that for most
of the OTUs, either there is not sufficient resolution at the 16S rDNA level to detect it, or, more
parsimoniously, these OTUs represent taxa that can colonize various substrates. Many in this group of
OTUs showing no preference may be adventitious endoliths that do not bear the burden of boring into
the substrate and can potentially colonize any substrate, but at least some represent most likely
euendoliths (NR_OTU4, OTU 351529 and OTU 842393), and still they do not seem to show
preference at this level of genetic resolution.

Using the same method, we then compared Ca-carbonate dominated samples (n=14) to Ca-Phosphate
dominated samples (n=3). The paucity of phosphate samples certainly restricted our statistical power,
but even then we were able to identify 81 OTUs that were statistically significantly enriched on the
phosphatic substrate ($p<0.05$) side, while only 21 were enriched in carbonates ($p<0.05$) (Figure 5). This
suggests an asymmetrical effect of carbonate vs. phosphate substrate types, the latter being a more
powerful driver of differential abundance among cyanobacteria. But again, in this case, the majority of
OTUs, including some of the most abundant, were promiscuous. *Mastigocoleus sp.* (NR_OTU193)
appeared clearly enriched in the carbonates ($\log_2$ fold difference = |3.8|), while the other potential
borers including the Pleurocapsales OTUs did not exhibit statistically significant substrate preference.

In all, these results suggest that some cyanobacteria do have a substrate preference, and that these
preferences sometimes occur among closely related clades (like NR_OTU193 and NR_OTU4), which
do exhibit differential occurrence. These comparisons highlight the differential role of the cationic *vs.*
the anionic mineral component. NR_OTU193 for instance showed a preference for both components, it
prefers calcium over magnesium in terms of cation and carbonate over phosphate as an anion. On the
other hand, NR_OTU741 only appeared differentially represented when the cationic part of the mineral





varied. Finally, it is important to note that only a small fraction of the cyanobacterial community seems
to be influenced by the substrate, 3.5% of the total number of species on average. These results are
consistent with the idea that borers may be specialized, but ancillary endoliths are not. The substrate
specialization of euendoliths may be due to the physiological requirements of excavation into specific
mineral types. Future endolithic community metagenomic reconstructions and comparisons could aid in
the identification of alternative pumps that may be specific to mineral types.

*Implications for the diversity of the boring mechanism and substrate-driven evolution of euendoliths*

A question that follows naturally from the previous findings is how such a substrate preference may
relate to the physiological mechanism of boring. The model strain *Mastigocoleus testarum* BC008 is
clearly specialized in the excavation of calcium carbonate through the uptake of calcium anions at the
boring front and their active transport along the filament toward the surface (Garcia-Pichel et al., 2010;
Guida and Garcia-Pichel, 2016). In culture, *M. testarum* strain BC008 could not bore into dolomite or
magnesite. In agreement with this, the closest phylogenetic allies to this strain in our communities,
(NR_OTU193) did also show a preference for calcium carbonates over magnesium carbonate.
Experiments with natural endolithic communities using calcium pump inhibitors have shown that the
calcium-based mechanism is commonly at work in many localities but, at least in one case, boring was
impervious to inhibition, pointing to the potential existence of mechanistic diversity (Ramírez-Reinat
and Garcia-Pichel, 2012b). Because we could not detect preferential enrichment of *bona fide*
euendoliths in the phosphate compared to the carbonate substrates, we must assume that the mineral
anion is not a strong determinant of substrate choice in these communities. The boring mechanism
described for *M. testarum BC008* is in fact only dependent on the nature of the cation, and could work
in principle on calcium phosphates as well, and yet *M. testarum* strain BC008 did not bore into pure
hydroxyapatite in the laboratory. These contrasted findings highlight that there must be factors other




than the cationic part of the mineral determining the excavation ability of a particular strain and that the
boring mechanism proposed for *M. testarum* strain BC008 might be only incompletely described.

**Conclusion**

An in depth survey of endolithic microbial communities associated to Isla de Mona intertidal outcrops
revealed a high diversity of organisms, comparable to those one found in other benthic marine
microbial communities such as the intertidal sediments and rock surfaces. These complex communities
likely host various microbial metabolic guilds beyond oxygenic phototrophs described during more
than a century of naturalist's descriptions. The analysis of the cyanobacterial community revealed the
prominence of possible euendolithic species belonging to all the known microborers genera except
perhaps *Plectonema*. Contrasting with results obtained at higher taxonomical level, substrate preference
could only be detected among cyanobacteria at the OTU level and close relatives have different
distribution patterns, arguing for the existence of boring mechanisms somewhat different to the one
described in the model strain *Mastigocoleus testarum*.

**Acknowledgment**
The authors would like to thank the Goldwater Materials Science Facility for their support in sample
preparation and analysis. The authors would like to acknowledge Christophe Thomazo for his
contribution to the "Euendolight" project and Purificación López-García for providing the Alchichica
samples.

**Authors contribution:** F. G.-P. and E.C. designed the experiment. F. G.-P., D.R., B.S.G. performed the
field work. The experimental work was done by D.R. and E.C. E.C. analyzed the results. and E.C. and
F. G.-P. prepared the manuscript with contribution from all co-authors.



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



**Figures Captions**

**Figure 1: Isla de Mona setting** (a) Simplified geological map modified from that of (Briggs and Seiders, 1972) showing the locations of the sampling sites. (b) Sky view of Isla de Mona, the cliff is composed of the Isla de Mona Dolomite topped by the Lirio limestone, the Isla de Mona lighthouse is visible (c-d) Views of Isla de Mona coastal area, samples were taken from isolated boulders (c), directly from the cliff (d) at the notch (white arrows c-d) or on the raised reef flat (c-d).

**Figure 2: Mineral composition and microbial community structure of Isla de Mona intertidal outcrops** Each line corresponds to one sample. (a) Mineralogical composition as retrieved by bulk powder XRD (b) Distribution of 16 rDNA OTUs taxonomically assigned at the phylum level and associated chao1 richness metric (c). This reflect the total microbial community structure (d) Distribution of the cyanobacterial 16 rDNA OTUs assigned at the phylum level, excluding chloroplasts and associated chao1 richness metric for Cyanobacteria (e).

**Figure 3: Hierarchical clustering analysis (UPGMA) of bacterial community composition in various settings based on pairwise Bray Curtis distance metrics.** The robustness of the topology was assessed through jackknife repeated resampling of 5000 sequences. The number of samples in a given collapsed tree branch are in parentheses, while the numbers in brackets are the Qiita dataset ID number.

**Figure 4: Differential abundance of cyanobacterial OTUs in Ca-carbonates (calcite-aragonite)**



**n=14 vs. CaMg-carbonate (dolomite) n=13 samples.** This plot was constructed using the DESeq2
method. It displays the average normalized counts per OTU as a measure of abundance against the log2
fold difference. The OTUs that were significantly differentially abundant in the two conditions
(p<0.05) are represented as open circles, the other ones are displayed as close symbols. Positive values
indicate enrichment towards CaMg-carbonate and negative values indicate enrichment towards Ca-
Carbonate. The OTU ID and taxonomical assignment of the most abundant OTUs is displayed on the
right. The stars tag the possible euendolithic OTUs as determined by phylogenetic proximity to known
microborers (Figure S3).


**Figure 5: Differential abundance of cyanobacterial OTUs in Ca-carbonate (calcite-aragonite)**
**n=14 vs. Ca-phosphate (apatite) n=3 samples** This plot was constructed using the DESeq2 method. It
displays the average normalized counts per OTU as a measure of abundance against the log2 fold
difference. The OTUs that were significantly differentially abundant in the two conditions (p<0.05) are
represented as open circles, the other ones are displayed as close symbols. Positive values indicate
enrichment towards Ca-phosphate and negative values indicate enrichment towards Ca-Carbonate. The
OTU ID and taxonomical assignment of the most abundant OTUs is displayed on the right. The stars
tag the possible euendolithic OTUs as determined by phylogenetic proximity to known microborers
(Figure S3).







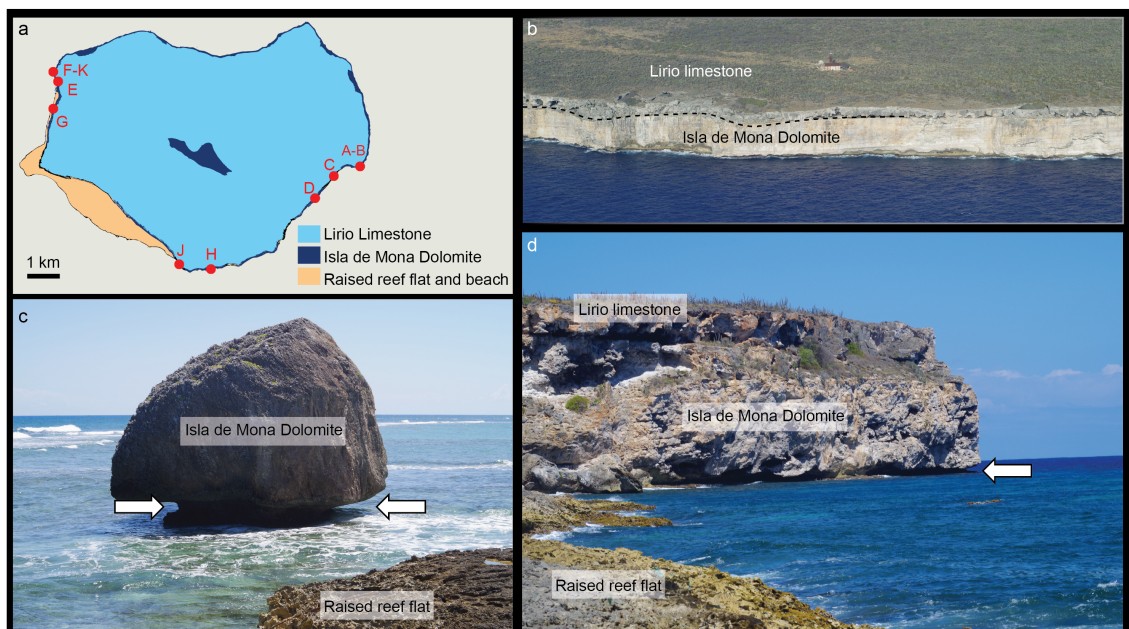

**Figure 1**



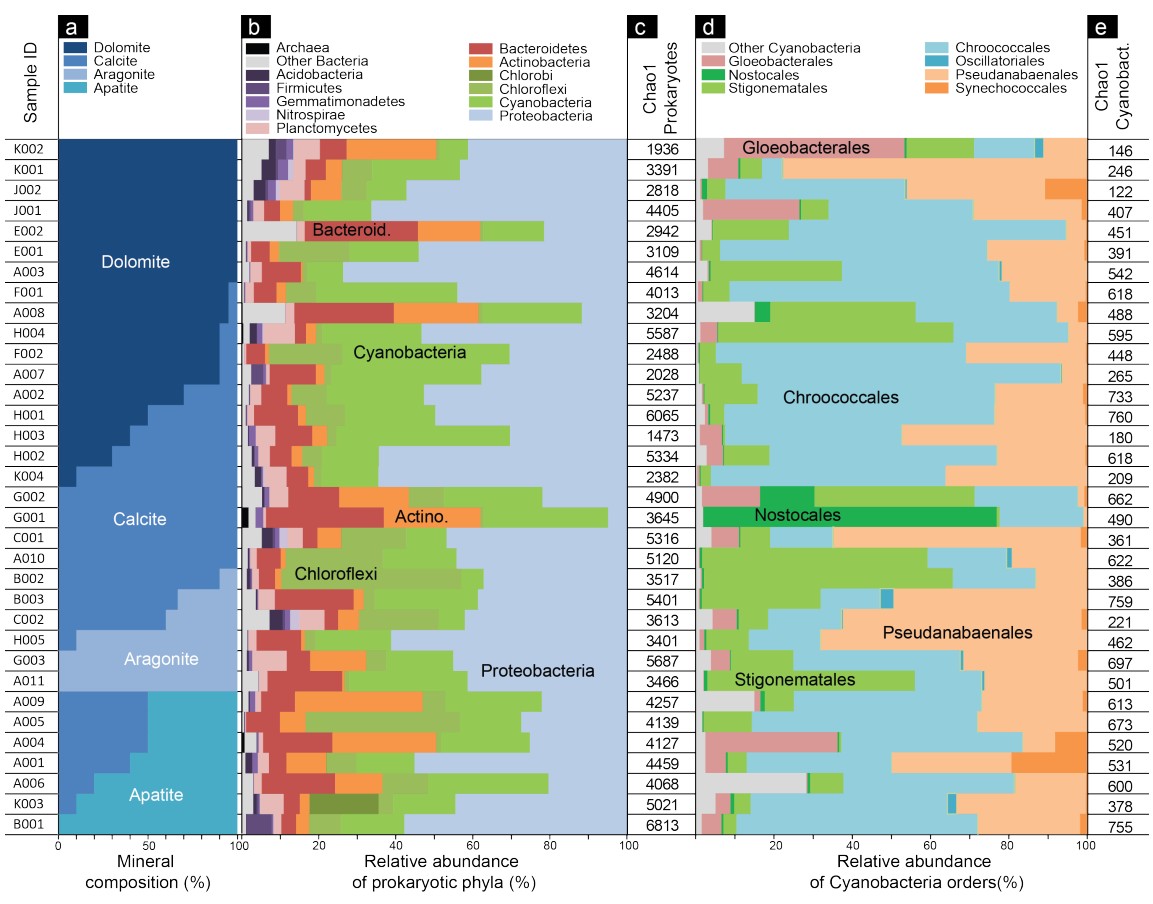


**Figure 2**





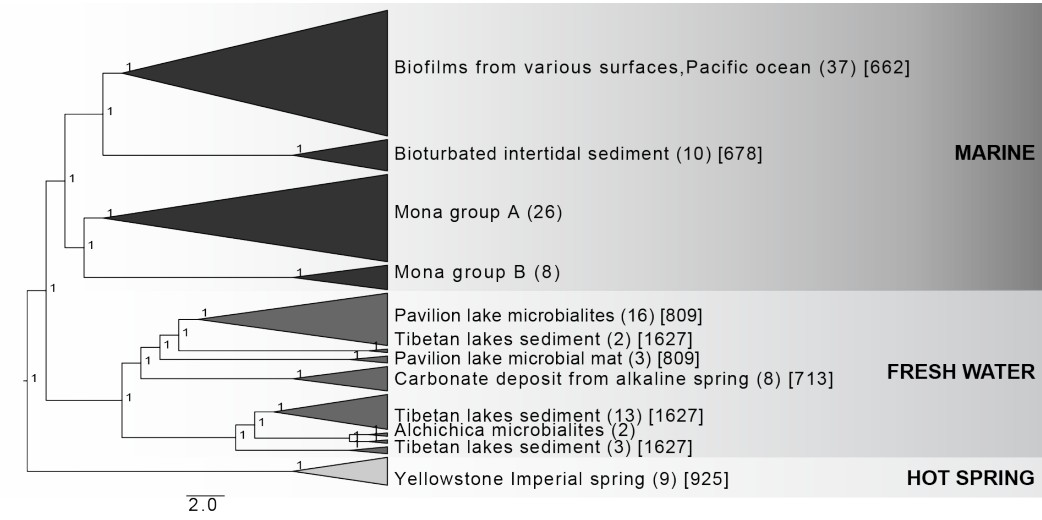


**Figure 3**



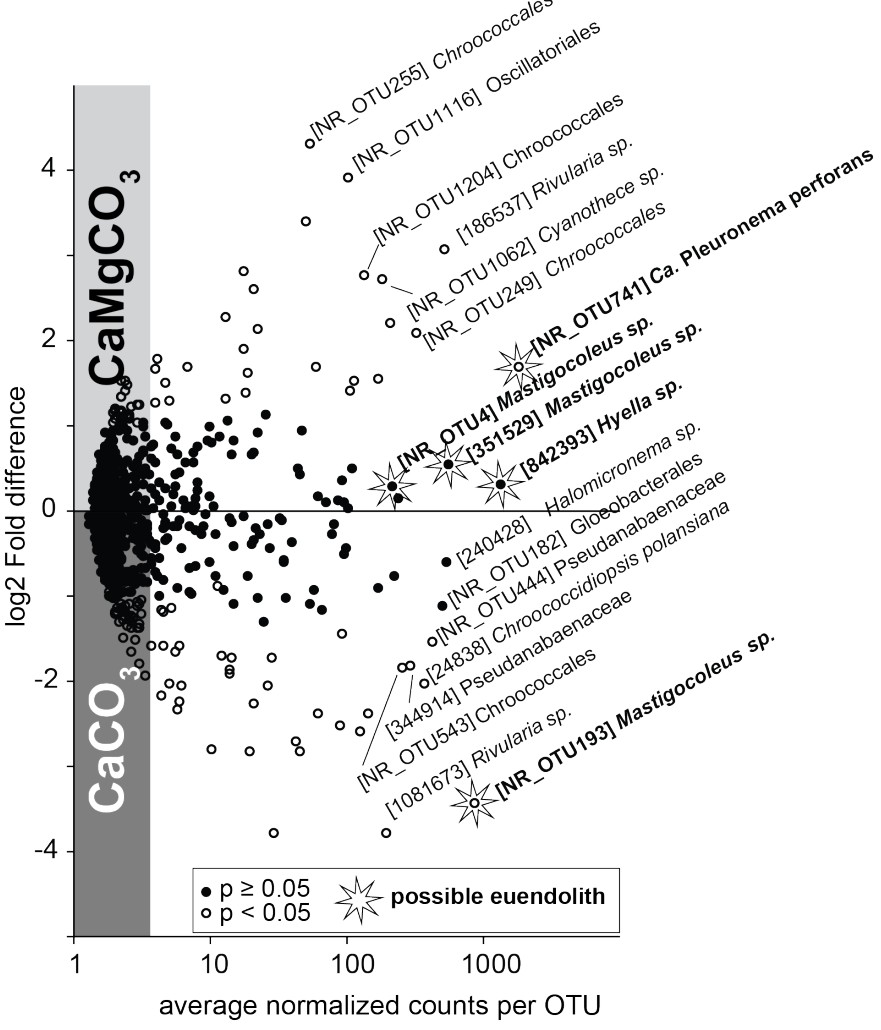


**Figure 4**






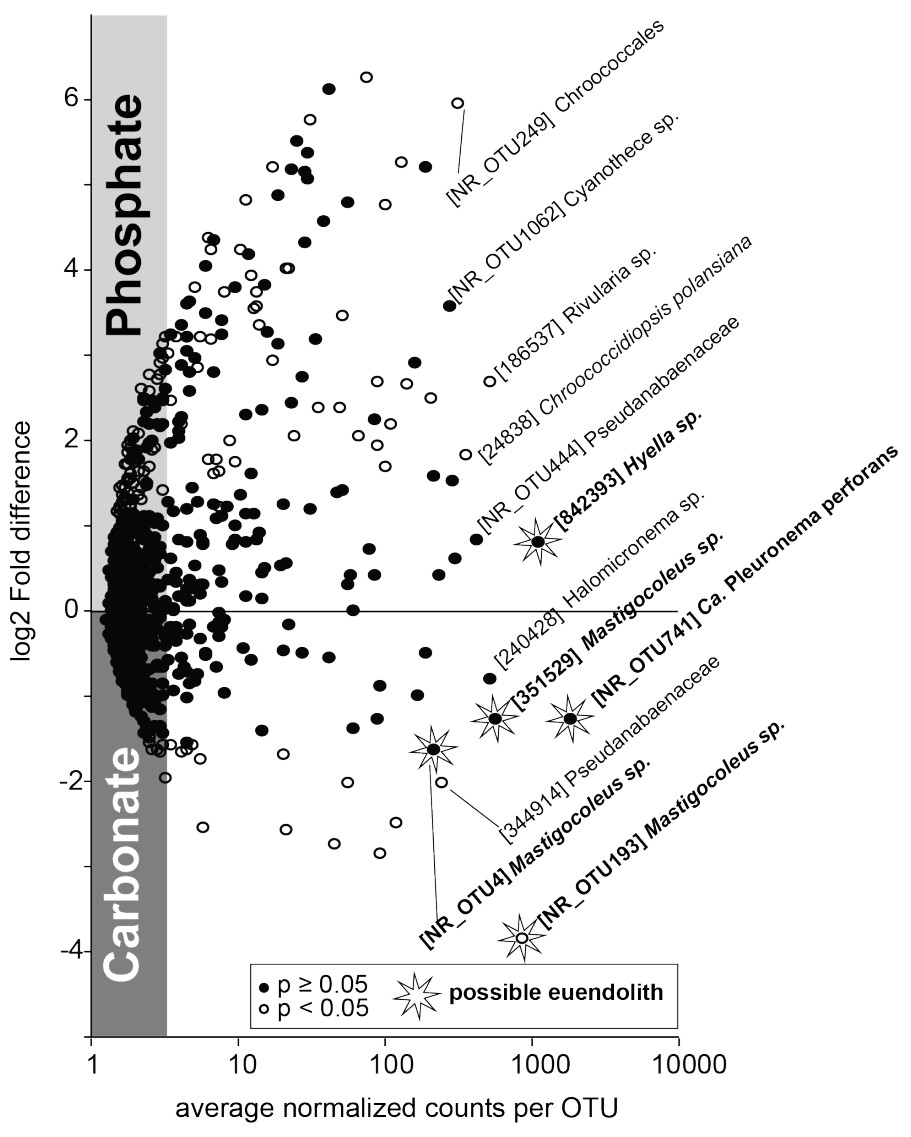


**Figure 5**




**Table 1: Euendolithic cyanobacterial strains used to assign potential roles to OTUs**

| Strain name | order | reference sequence | presence in this dataset | Isolation source | bores in culture | reference |
|---|---|---|---|---|---|---|
| *Mastigocoleus testarum* | Stigonematales | DQ380405 | yes | Cabo Rojo carbonate, Puerto Rico | yes | (Chacón et al., 2006) |
| *Solentia sp. HBC10* | Pleurocapsales | EU249126 | no | Stromatolite bahamas | yes | (Foster et al., 2009) |
| *Hyella sp. LEGE 07179* | Pleurocapsales | HQ832901 | yes | Rocky Moledo do Minho beach (Portugal) | not tested* | (Brito et al., 2012) |
| *Ca.* **Pleuronema perforans IdMA4** | Pleurocapsales | KX388631 | yes | Isla de Mona outcrop | yes | *this study* |
| *Ca.* **Mastigocoleus perforans IdM** | Stigonematales | KX388632 | yes | Isla de Mona outcrop | yes | *this study* |
| *Ca.* **Pleuronema testarumRPB** | Pleurocapsales | KX388633 | Yes | Puerto Peñasco Coquina reef | yes | *this study* |

*Hyella sp. LEGE 07179 was isolated from inside a patella shell where it was identified as a true borer by the
authors but its boring ability was never tested again in the lab