# Peer review of "Diversity and mineral substrate preference in endolithic microbial communities from marine intertidal outcrops (Isla de Mona, Puerto Rico)."

_Biogeosciences, 2016_

## Referee Comment (RC1) · Anonymous Referee #1 · 25 Aug 2016

Diversity and mineral substrate preference in endolithic microbial communities from marine intertidal outcrops (Isla de Mona, Puerto Rico).

Garcia-Pichel et al.

This study investigated substrate specificity of endolithic communities in phosphate rock, limestone and dolostone outcrops from Isla de Mona, Puerto Rico. Authors implemented a high-throughput 16SrDNA genetic diversity approach that reveled the dominance of euendolithic cyanobacteria associated to a high community diversity. Results did not support the hypothesis that community composition would relate to mineral substrate but particular euendolithic cyanobacteria seemed to be specialized at the mineral substrate level. Also, the question regarding the existence of a specialized community

associated to dolostone vs limestone could not be resolved. Since authors used a very short region of the 16S rDNA and then used a culture-collection of euendoliths to extract the euendolith sequences, there seems to be a lot diversity that was not included, thus they conclude that only a small fraction of the community (3.5%) is influenced by substrate. The data analysis need to be redone. The method used provides short fragments and does not allow for thorough phylogenetic analysis. This study would greatly benefit from longer reads and maybe a metagenomic approach.

The main question addressed was if there is a highly adapted endolithic flora to specific mineral substrates, yet in lines 299-301, authors state "At this level of taxonomic resolution, we did not detect any significant association of substrate mineralogy and community composition". To answer the main question authors used high-throughput sequencing to describe the microbial diversity and test the effect of different substrates on community composition.

Specific comments: Abstract- The last claim "The cationic mineral component was…. existence in nature of alternatives to the boring mechanism…. based…on transcellular calcium transport" is not sustained from the results presented.

Methods- Authors justify the need to re-assess the diversity of euendolithic cyanobacteria yet only include a high-throughput sequencing approach that produced very short reads, which are not informative for phylogenetic analysis. Itag sequencing is not the best platform to analyze deep phylogenetic affiliations and to resolve the mentioned issues on euendolithic cyanobacteria for this study model.

Lines 196-201, repeated phrase. Line2 209-212, I don't understand why mention another site, and sequences that are afterwards not discussed in this analysis. Lines 237-238, by using a dataset with proven boring cultured strains and using that to assess which of the cyanobacteria OTUs could be euendolyths, this study is losing the power to identify other euendoliths. Why compare only to the known euendolith dataset?

Results and Discussion Lines 274-275, please give information on coverage.

Lines 307-314, in this paragrapha, authors mention that the sequences obtained in this study clustered together and discuss that 1) this could happen since euendolithic assemblages are distinct in a microbiological and adaptive way, or that alternatively 2) the clustering pattern reflects a biogeographical island effect, since all samples come from a small area. Authors discuss the second is unlikely given the cosmopolitan nature of marine cyanobacteria. Nonetheless the references cited are for a cosmopolitan, nonmarine cyanobacterium, M. vaginatus and for M. chthonoplastes. This discussion should be revised; there are different methods to proof for biogeography in communities, and to analyze diversity patterns related to biogeography. Also, it is possible to do analysis to disentangle which environmental variables, in this case including mineral composition, are relevant and explain community composition.

Line 323, are there cyanobacterial communities? Or populations that interact with others to form communities? Line 327, The cyanobacterial community (diversity?) appeared quire diverse (elevated?) with a specific. . ..

Lines 348-349, What percentage of the community to euendolithis represent?

Lines 353-355, Authors could do microscopic observations to make sure the issue regarding the lack of P. terebrans.

Lines 451-458, This discussion s very interesting bot out of place since this study did not focus on M. testarum but on the overall boarer diversity.

Conclusion- Lines 462-466, please revise use of English. "These complex communities likely host. . . This phrase is stating the obvious. Lines 468-471, the claim regarding different boring mechanisms than those known for M. testarum is not sustained from these results.

---

## Referee Comment (RC2) · Anonymous Referee #2 · 24 Sep 2016

Nice study overall, and a valuable contribution.

Some edits suggested for clarification.

A concern, which can be addressed, is over use of the term "preference" indicating observation of higher density of colonization on some substrates, being over interpreted to prove preferences for specific cations or anions.

There are many variable factors in a natural environment. First looking at distribution in a quantified manner allows hypotheses for testing to illustrate actual "preference"

Please also note the supplement to this comment:

[Figure]

http://www.biogeosciences-discuss.net/bg-2016-254/bg-2016-254-RC2-supplement.pdf

[Figure]

**Supplement:**

bg-2016-254 Edit Suggestions and comments

Line 21: define "dominant" as compared to what?  Phototrophs? OTUs? Biomass

Line 27:  substitute "suggesting" for "calling for"

Line 44: MacIntyre et al, describe a two phase boring and subsequent new crystal formation within boreholes (within grains and connecting to adjacent) rather than external cementation of grains.

Line 48: m-2 d-1 and are of clear

Line 73: end sentence at "including genetic markers." Delete (a task yet to be undertaken with any breadth) or re-word at the beginning and add "To date" these genera were all assigned….

Line 77:  suggest deleting "alternative" and edit to  "complimentary and more comprehensive descriptions" of endolithic….

Line 79: delete "merely" and "morphological studies alone"

Line 81: delete "brought to our attention" substitute "have revealed"

Line 82: delete "not just composed of" substitute "in addition to"

Comment: this section on associations and succession patterns is well placed

Line 90:  Suggest re-wording "However similar studies using the power of high throughput sequencing techniques are not yet available for the globally….."

Line 100: delete "could show" substitute  we "found"

Line 101: delete "fastest" substitute "most rapidly"

Line 106: delete "arguments" suggest rewording "Similar substrate preferences have also been observed for phosphates"

Comment:  lines 110 to 115, intriguing and compelling questions.

Line 112: Suggest re-wording.  Perhaps, "We therefore developed the line of inquiry to ascertain if evolutionary specialization has resulted in a highly adapted….

Line 114: Suggest deleting sentence beginning with "Surprisingly, this aspect…."

The previous line is stronger without Line 114.

Comment: Lines 117 to 123, nicely and succinctly phrased.

Question, line 135: Was the geologic hammer sterilized before sampling? - See that sterilization steps were taken in the lab in line 157.

Line 137 and paragraph: seems awkward and a little confusing. What were replicates?

Perhaps "At each sample location three replicate aliquots of rock chips were collection in sterile 50 mL falcon tubes."

Presumably Air drying was done in the lab? And preservation in alcohol in the field?

Text should clarify this here. Also please specify how long in transit and transit conditions (light/ dark?/ .

Where were the seawater aliquots collected relative to the sample locations? (add to map in Fig 1.) and specify how handled/ stored and time to analysis.

Line 149: Could delete the word "retrieve"

Line 158: A chip was further "ground" not "grounded"

Line 161: was modified "by homogenizing bead tubes" delete "as follow" and delete "were applied"

Line 176: "were performed"

Line 182: "barcode" removal

Line 184: processed

Line 188: "further report specific abundances for each sample"

Line 189: Suggest "Because this study focused on the most abundant OTUs and how they vary, rather than the rare biosphere, we filtered….to remove the few? rare OTUs"

Question how many rare OTUs were present?

Line 207: Suggest, "For comparison, raw sequences….

Line 212: delete "they were" substitute "and also" delete "as well"

Line 222: "ran"

Line 254-256: worded a little awkwardly,

Line 290: communities "in this study" are much more complex than the majority of literature "to date"

Line 294: proven that "some axenic" cyanobacteria delete "alone"

Line 296: other metabolic activities (of other co-occurring microorganisms) particularly those that result in pH changes….

Comment Lines 307-320 Nice discussion

Line 349: delete "does" ie., "one that not only represents" an initial set of pioneers…

Line 356: delete "always" substitute (has "so often" been described…)

And delete "who can" substitute "accounting" ie., "community accounting for"

Line 360. "were" instead of "was"

Question: are attempts in progress to isolate *Plectonema terebrans*? And are isolates available from other sources?

Minor editing lines 385 to 403

Line 404 delete "It becomes clear that substrate preferences" suggest substitute "Results suggest substrate preferences are found…"

Some rewording of section line 410 to 414, probably split into two sentences

Line 417 suggest "although the paucity of samples restricted our statistical power, we were still able to identify…"

Line 422: suggest delete "promiscuous" which is vague substitute "widespread across different substrate types"

Comment: At the beginning of the manuscript "substrate preference" refers to clear numerical/ statistical occurrence (of particular endolithic cyanobacterial species relative to other species) in different mineral/ rock types. However at this point in the manuscript the term is "preference" is seeming to take on a more determinative meaning that is not yet demonstrated. Preference can be a tricky term to use. implying a more "decision based"

A particular species may seem to show a preference for a Mg or Ca cation containing mineral substrate based on occurrence/ density, but that does not necessarily imply a "preference" for Mg or Ca cations. That would need to be tested independently, as would the "preference" for anions.

The authors seem to understand this, but still sometimes fall into an overly interpretive phrasing implying metabolic/behavior from a distribution.

Certainly the data suggest some trends worth rigorously testing, (as has been done for *M. testarum* BC008) and it is critical to begin the determination of substrate preferences, by detailed investigations of naturally occurring distributions as the authors have done.

Line 447: in light of previous discussion would suggest changing "preference" to

"NR_OTU193) did show a higher rate of occurrence  in calcium carbonates as compared to magnesium carbonate.

A careful re-reading and edits with particular emphasis on the implications and possible over interpretation of the term "preference" would be very beneficial.

Line 452:  delete "we must assume" again data "suggests" but does not "prove"

Comment:   Authors conclude that more factors may be involved in substrate preference that cation preference alone.  A bit more discussion of what those other factors may be would be helpful.

Perhaps also in the introduction a short synopsis on thought / previous research as to "why" boring behavior is prevalent in some groups would be informative (ie., is behavior thought to provide protection from wave energy/ nutrients/cations/ light modification)?

---

## Author Comment (AC1) · 10 Oct 2016

Taxonomic note

Due to conflicting taxonomy (see NCBI taxonomists report below) , the genus name "Pleuronema" will be replaced by the name "Pleurinema" throughout the manuscript. The accession numbers remain unchanged.

NCBI taxonomists:

The use of the genus name 'Pleuronema' for cyanobacteria is illegitimate as it is already in use for a genus of Ciliophora (Pleuronema Dujardin, 1841). You should modify the associated names in your submission (and in your manuscript) accordingly.

---

## Author Comment (AC2) · 10 Oct 2016

Diversity and mineral substrate preference in endolithic microbial communities from marine intertidal outcrops (Isla de Mona, Puerto Rico).

Garcia-Pichel et al.

> We would like to thank the referee #1 for the time spent in a thorough review and his/her comments. We take this opportunity to address his/her concerns regarding the approach and methodology choices that we made. Before that we would like to set for the record straight that Garcia-Pichel is the senior author, not the lead, of this article, therefore usage would indicate to refer to this contribution as Couradeau et al.

This study investigated substrate specificity of endolithic communities in phosphate rock, limestone and dolostone outcrops from Isla de Mona, Puerto Rico. Authors implemented a high-throughput 16SrDNA genetic diversity approach that reveled the dominance of euendolithic cyanobacteria associated to a high community diversity. Results did not support the hypothesis that community composition would relate to mineral substrate but particular euendolithic cyanobacteria seemed to be specialized at the mineral substrate level. Also, the question regarding the existence of a specialized community associated to dolostone vs limestone could not be resolved.

Since authors used a very short region of the 16S rDNA and then used a culture-collection of euendoliths to ex- tract the euendolith sequences, there seems to be a lot diversity that was not included, thus they conclude that only a small fraction of the community (3.5%) is influenced by substrate. The data analysis need to be redone. The method used provides short frag- ments and does not allow for thorough phylogenetic analysis. This study would greatly benefit from longer reads and maybe a metagenomic approach.

> We, as referee #1, do principally worry about technical aspects, but have to disagree with the criticisms leveraged at our approach and analyses.
>
> Being aware that the length of the sequences is critical for phylogenetic reconstruction, we took advantage of the recent progress of the Illumina chemistry and use general primers that amplify 465bp of the 16S (V3-V4 regions) instead of classical set of primers centered on the V3 region only (291bp) (Caporaso et al., 2012). To reconstruct the phylogeny (Figure S3) we manually selected 736 well aligned positions from our multiple alignment. The ends of our Illumina reads were filled up with the "?" character treated as an unknown position by the evolutionary model as defined in the Treefinder manual http://www.treefinder.de/tf-march2011-manual.pdf. The obtained topology is well supported and allows us to resolve the position of the OTUs of interest compared to reference sequences. We made the careful assumption that only OTUs that fell within clades of proven euendolithic strains could be deemed possibly euendolithic themselves. To do so we decided to put effort into increasing the number of reference sequences of proven euendolithic strains through targeted cultivation.
>
> We note that our sequencing and bioinformatics approach is currently standard in microbial ecology. Here are a few examples of recent papers that used the same technique to detect microbial dynamics and distribution:
>
> Angelakis, E., Yasir, M., Bachar, D., Azhar, E.I., Lagier, J.-C., Bibi, F., et al. (2016). Gut microbiome and dietary patterns in different Saudi populations and monkeys. Sci. Rep., 6, 32191
> Boetius, A., Anesio, A.M., Deming, J.W., Mikucki, J.A. & Rapp, J.Z. (2015). Microbial ecology of the cryosphere: sea ice and glacial habitats. Nat Rev Micro, 13, 677–690

Clayton, J.B., Vangay, P., Huang, H., Ward, T., Hillmann, B.M., Al-Ghalith, G.A., et al. (2016). Captivity humanizes the primate microbiome. Proc. Natl. Acad. Sci., 113, 201521835

Hu, J., Raikhel, V., Gopalakrishnan, K., Fernandez-Hernandez, H., Lambertini, L., Manservisi, F., et al. (2016). Effect of postnatal low-dose exposure to environmental chemicals on the gut microbiome in a rodent model. Microbiome, 4, 26

Lal, C.V., Travers, C., Aghai, Z.H., Eipers, P., Jilling, T., Halloran, B., et al. (2016). The Airway Microbiome at Birth. Sci. Rep., 6, 31023

Props, R., Kerckhof, F.-M., Rubbens, P., De Vrieze, J., Sanabria, E.H., Waegeman, W., et al. (2016). Absolute quantification of microbial taxon abundances. ISME J. Adv. online Publ., 1–4

The main question addressed was if there is a highly adapted endolithic flora to spe- cific mineral substrates, yet in lines 299-301, authors state "At this level of taxonomic resolution, we did not detect any significant association of substrate mineralogy and community composition". To answer the main question authors used high-throughput sequencing to describe the microbial diversity and test the effect of different substrates on community composition.

We disagree on the reading of the results by the reviewer. This particular study had a double aim, (i) we wanted to apply the widely used 16 rRNA gene high throughput sequencing tool to describe intertidal endolithic communities and (ii) to test whether there exists a specialized community associated to the type of mineral they colonize. The motivation of our first aim was the lack of such dataset for these globally relevant microbial communities. The work presented here definitely contributes that part. The second aim was driven by the hypothesis that, if there exists a substrate preference of the pioneer euendolithic cyanobacteria, this preference could drive the total microbial community towards different climax communities.

Our dataset revealed that endolithic habitat hosts a large variety of microbial species, a lot wider that could have been foreseen from classical literature. As noted by the referee #1 we did not observe a correlation between the proportion of prokaryotic phyla and the mineralogy of their substrate. In other words, if there is a substrate preference it does not reflect into the proportion of prokaryotic phyla.

However, the proportion of a given phylum does not indicate the nature of the microbes that constitute it, therefore is not sufficient to reject our hypothesis. We demonstrated that there is substrate preference among the Cyanobacteria, so even if their proportion of the total community does not vary significantly with the substrate, their composition does. This clearly supports our hypothesis in a statistically robust and significant way. We could identify several cyanobacterial OTUs that were differentially abundant on limestone compared to dolostone (Figure 4). It is correct, as referee#1 pointed out, that only a small fraction of the cyanobacterial OTUs diversity (3.5%) showed a significant change in abundance with substrate. However, these very OTUs account for 16 ±4% of the total number of cyanobacterial sequences analyzed here (some of these OTUs, especially the possible euendoliths, being very abundant). Thus, they are not only differentially distributed, but also a significant proportion of the community.

We demonstrated that the effect of the substrate is not dramatic enough to change the proportion of prokaryotic phyla but still affects the abundance of some keystones species such as pioneer euendolithic Cyanobacteria.

We do agree with referee #1 that we could not bring a "yes / no" answer to the hypothesis; we argue that we enhanced our hypothesis by showing that the answer depends on the taxonomic level set for the analysis. Yes at fine resolution, no at coarse resolution.

We are grateful that referee #1 pointed out the reference sequences of the newly cultured euendolithic strains, as we realized that the details regarding the amplification and sequencing of their 16S rRNA genes were missing. We used the primers and PCR conditions recommended by Nübel to retrieve these sequences (Nübel et al., 1997). These details will be added to the methods section of the revised manuscript.

We agree with the referee #1 that a metagenomics study could be a nice follow-up step to the present piece of work. This contribution constitutes a pioneer study that explored the endolithic microbial diversity associated to various substrates. For that purpose, we used the 16S rRNA gene as a proxy that allowed us to genetically sample a large variety of locations with the appropriate sequencing depth. This is a required first step to ask relevant functional questions that could justify a new study involving metagenomics or other relevant methods such as in-situ biogeochemistry measurement, fluorescent labelling, and metabolomics.

Specific comments: Abstract- The last claim "The cationic mineral component was. . ... existence in nature of alternatives to the boring mechanism. . .. based. . .on transcellu- lar calcium transport" is not sustained from the results presented.

In their recent contribution, (Guida and Garcia-Pichel, 2016) showed that the boring mechanism in the model strain *Mastigocoleus testarum* BC008 was based on vectorial transcellular transport of calcium from the boring front to the boring hole. Here we show that some possible euendoliths, including close relatives to *Mastigocoleus testarum* BC008, do not show exclusive preference for ca-carbonate substrate. This indicates that the *Mastigocoleus testarum* BC008 vectorial transport of calcium ions to bore cannot be the sole mechanism, and that there might exist alternative mechanisms.

Methods- Authors justify the need to re-assess the diversity of euendolithic cyanobac- teria yet only include a high-throughput sequencing approach that produced very short reads, which are not informative for pylogenetic analysis. Itag sequencing is not the best platform to analyze deep phylogenetic affiliations and to resolve the mentioned issues on euendolithic cyanobacteria for this study model.

Our aim was to and compare the microbial diversity among 34 samples, for which we used 16S rRNA gene genetic sampling. Saying that a) read are shorts and b) uninformative, is simply incorrect. Again we point the reviewer to the fact that this is a standard methodology with the power to show differences (see some other examples above). The reads produced here were 465bp. Using 16S rDNA to assess the microbial diversity allowed us to both reach enough sequencing depth to get appropriate coverage (see Table S2) and to compare our sequences with the largest library of taxonomically assigned sequences (Greengenes 13-8).

Lines 196-201, repeated phrase.

This will be fixed.

Line2 209-212, I don't understand why mention an- other site, and sequences that are afterwards not discussed in this analysis.

We regret that our point was missed by the reviewer. We mentioned these samples because we included them in the meta-analysis, figure 4. We included them for comparative purposes, processing them in parallel to the Mona samples. They came from a different type of environment (alkaline lake) and therefore represent an internal control of our analysis to supports the fact that the difference that we see is due to the environment, rather than analytical. Differences due to analytical aspects are principally possible when comparisons are done on dataset retrieved from the Qiita database.

Lines 237- 238, by using a dataset with proven boring cultured strains and using that to assess which of the cyanobacteria OTUs could be euendolyths, this study is losing the power to identify other euendoliths. Why compare only to the known euendolith dataset?

How does one know the metabolic activity of any one particular organism identified based solely on the presence of its 16S rDNA? The best approach one can take is to compare this particular 16S rDNA to the reference sequences of organisms with proven activity. There is no other way we could have identified euendoliths, and one can never identify a new euendolith (or any other putative metabolic activity) based on a sequence only. This also justifies why we put effort into increasing the number of references sequences through targeted cultivation effort.

Again this approach, which is rather commonplace and the basis of most functional bioinformatics, will indeed miss absolute novelty, but will secure identification of a large part of the community.

Results and Discussion Lines 274-275, please give information on coverage.

Please see Table S2 column 2 for coverage information.

Lines 307-314, in this paragrapha, authors mention that the sequences obtained in this study clustered together and discuss that 1) this could happen since euendolithic assemblages are distinct in a microbiological and adaptive way, or that alternatively 2) the clustering pattern reflects a biogeographical island effect, since all samples come from a small area. Authors discuss the second is unlikely given the cosmopolitan na- ture of marine cyanobacteria. Nonetheless the references cited are for a cosmopolitan, nonmarine cyanobacterium, M. vaginatus and for M. chthonoplastes. This discussion should be revised; there are different methods to proof for biogeography in communi- ties, and to analyze diversity patterns related to biogeography. Also, it is possible to do analysis to disentangle which environmental variables, in this case including mineral composition, are relevant and explain community composition.

We would like to thank referee #1 for pointing us to a follow-up hypothesis that could be tested in the framework of this experiment. However, the point of the present contribution is not to discuss the biogeography of Cyanobacteria in general, but to look at substrate preference among endolithic communities. The meta-analysis that was performed here used an aggregated dataset from various studies looking at marine and lake sediments, intertidal mollusks shells, microbialites and hot springs.

We agree with referee #1 that it would be great to be able to correlate the pattern that we observed with some environmental parameters, however this type of data is not consistently available for the chosen datasets. We agree with the reviewer that this point of discussion being speculative it would be best to let our readers make their own opinion, we will therefore present the two hypothesis as equally valuable in the revised version of the manuscript.

> The references associated to that paragraph point both to *Microcoleus chtonoplastes* as an example of marine cosmopolitan cyanobacterium and not to *Microcoleus vaginatus*, a terrestrial biocrust forming cyanobacterium.

Line 323, are there cyanobacterial communities? Or populations that interact with others to form communities? Line 327, The cyanobacterial community (diversity?) ap- peared quire diverse (elevated?) with a specific. . ..

> We would like to thank referee #1 for pointing out some terminology ambiguities. Here we use the word "community" as an aggregation of all the cyanobacteria sequences, there is therefore one community of Cyanobacteria in our dataset. We further give a quantification of the specific diversity/ richness of this community using the classical chao1 alpha-diversity metrics. We avoided the term "population" as this term might refer to population genetics and within species interaction which was not the subject of the present contribution.

Lines 348-349, What percentage of the community to euendolithis represent?

> Good point. Euendoliths represent (based on the 7 OTUs that we could assign as putative euendolith based on their phylogenetic proximity to known microborers only) from 0.8% to 73% of the sequences depending of the sample considered. (Average value 29%). We will include this relevant information in the revised manuscript.

Lines 353-355, Authors could do microscopic observations to make sure the issue regarding the lack of P. terebrans.

> Microscopy will be insufficient. Referee #1 will agree that it is particularly challenging to recognize *Plectonema terebrans*, this species being described based on very common morphological characteristics:
>
> *"Fila gracilia, elongata, flexuosa, vulgo parce pseudo-ramosa, pseudo-ramis saepius solitariis. Vaginae hyalinae tenuissimae , cylindraceae , chlorozincico iodurato non caerulescenles. Trichomata dilute aeruginea, non torulosa, 0,95 µ ad 1,5 µ crassa; arliculi diametro trichomatis longiores, 2 µ ad 6 µ longi; dissepimenta binis granulis protoplasmaticis nolata; cellula apicalis rotundata (v. v.)"* Gomont, M. (1892 '1893'). Monographie des Oscillariées.
>
> In fact, this description would even fit well members of the Chloroflexus bacteria, which are also present. Therefore, it is possible that several species, even non boring colonies that were secondary colonizers, have been called *P. terebrans* over the years, genetic tools would in that case help to resolve the abundance/presence of a particular boring *P.terebrans*, unfortunately there is no available reference sequences for this group. We hope that a cultured isolate will provide a reference sequence for *P.terebrans* in the future to help us overcome the limitations of microscopic observations for this group, similar to what we did for other euendolithic clades in the present contribution.

Lines 451-458, This discussion s very interesting bot out of place since this study did not focus on M. testarum but on the overall boarer diversity.

> This part of the discussion focuses on how the current findings contrast with the knowledge that was gained from the only model strain of euendolithic cyanobacteria that exists, *M. testarum* BC008. We

regret that referee #1 judged it out of place as it seems important to tell our reader how these novel findings relate with the proposed boring mechanism deciphered from physiological studies of *M. testarum* BC008. So far, this strain is the single model one can compare with, so certainly not out of place.

Conclusion- Lines 462-466, please revise use of English. "These complex communities likely host. . . This phrase is stating the obvious.

*"These complex communities likely host various microbial metabolic guilds beyond oxygenic phototrophs described during more than a century of naturalist's descriptions."*

This sentence is recapitulating an important finding of this study which is that these communities are more diverse and likely hold more metabolic capabilities than one could have expect from previous literature. This was not obvious until the entire community (including heterotrophic members) was described using 16S rDNA based genetic sampling of the community here.

Lines 468-471, the claim regarding different boring mechanisms than those known for M. testarum is not sustained from these results.

This discussion point aims at casting the results presented here in the framework of the model developed for *M. testarum* BC008. See answer to the first specific comment above for more details.

References cited in the answer to referee #1

Caporaso, J. G., Lauber, C. L., Walters, W. a, Berg-Lyons, D., Huntley, J., Fierer, N., Owens, S. M., Betley, J., Fraser, L., Bauer, M., Gormley, N., Gilbert, J. A., Smith, G. and Knight, R.: Ultra-high-throughput microbial community analysis on the Illumina HiSeq and MiSeq platforms., ISME J., 6(8), 1621–4, 2012.

Guida, B. S. and Garcia-Pichel, F.: Extreme cellular adaptations and cell differentiation required by a cyanobacterium for carbonate excavation, Proc. Natl. Acad. Sci., in press, 2016.

Nubel, U., GarciaPichel, F., Muyzer, G. and Garcia-pichel, F.: PCR Primers To Amplify 16S rRNA Genes from Cyanobacteria, Appl. Environ. Microbiol., 63(8), 3327–3332, 1997.

---

## Author Comment (AC3) · 10 Oct 2016

Referee#2

Nice study overall, and a valuable contribution.
Some edits suggested for clarification.

We would like to thank referee #2 for her/his thoughtful comments, and are grateful for the effort placed into reviewing in great detail the content and wording of our manuscript. We are pleased that she/he found out study nice and valuable and will address his/her comments and edits in detail below.

A concern, which can be addressed, is over use of the term "preference" indicating observation of higher density of colonization on some substrates, being over interpreted to prove preferences for specific cations or anions.
There are many variable factors in a natural environment. First looking at distribution in a quantified manner allows hypotheses for testing to illustrate actual "preference"

In hindsight, we agree that this term definition involves already a level of interpretation that is not adequate or useful to describe the results, we will therefore reserve that term for the discussion part.

bg-2016-254 Edit Suggestions and comments

Line 21: define "dominant" as compared to what? Phototrophs? OTUs? Biomass

Line 27: substitute "suggesting" for "calling for"

Line 44: MacIntyre et al, describe a two phase boring and subsequent new crystal formation within boreholes (within grains and connecting to adjacent) rather than external cementation of grains.

Line 48: m-2 d-1 and are of clear

Line 73: end sentence at "including genetic markers." Delete (a task yet to be undertaken with any breadth) or re-word at the beginning and add "To date" these genera were all assigned....

Line 77: suggest deleting "alternative" and edit to "complimentary and more comprehensive descriptions" of endolithic....

Line 79: delete "merely" and "morphological studies alone"

Line 81: delete "brought to our attention" substitute "have revealed"

Line 82: delete "not just composed of" substitute "in addition to"

Line 21: Here we meant "dominant" compared to

Line 27: We will do so in the revised version of the manuscript.

Line 44: *"the cementation of loosely bound carbonate grains in coastal stromatolites"* We would like to thank referee #2 for pointing out these details, we agree that the use of the term "cementation" implying the trapping of the grains by an external matrix does not describe accurately the process at play. We will rephrase this sentence as follow in the revised manuscript: *"the formation of lithified laminae of welded carbonate grains in coastal stromatolites"*

Line 48: ok

Line 73: We will add "To date"

Line 77: Good suggestion. We will replace "alternative" by "more comprehensive"

Line 79-82 We agree to these edits.

Comment: this section on associations and succession patterns is well placed

Line 90: Suggest re-wording "However similar studies using the power of high throughput sequencing techniques are not yet available for the globally….."

Line 100: delete "could show" substitute we "found"

Line 101: delete "fastest" substitute "most rapidly"

Line 106: delete "arguments" suggest rewording "Similar substrate preferences have also been observed for phosphates"

Comment: lines 110 to 115, intriguing and compelling questions.

Line 112: Suggest re-wording. Perhaps, "We therefore developed the line of inquiry to ascertain if evolutionary specialization has resulted in a highly adapted….

Line 114: Suggest deleting sentence beginning with "Surprisingly, this aspect…."

The previous line is stronger without Line 114.

Comment: Lines 117 to 123, nicely and succinctly phrased.

We are pleased that referee #2 found some sections well placed, well written or even compelling enough to be mentioned.

Line 90: We will reword this sentence as follow: ""*However, no high throughput sequencing studies are available on the globally significant intertidal endolithic communities.".*"

Line 100-101-106: We agree to these edits.

Line 112-114: We would like to keep these two sentences as they are because we think that it is important to state very directly the originality of the present study in the framework of the existing literature especially for a non-specialist audience.

Question, line 135: Was the geologic hammer sterilized before sampling? - See that sterilization steps were taken in the lab in line 157.

Line 137 and paragraph: seems awkward and a little confusing. What were replicates?

Perhaps "At each sample location three replicate aliquots of rock chips were collection in sterile 50 mL falcon tubes."

Presumably Air drying was done in the lab? And preservation in alcohol in the field?

Text should clarify this here. Also please specify how long in transit and transit conditions (light/ dark?/ .

Where were the seawater aliquots collected relative to the sample locations? (add to map in Fig 1.) and specify how handled/ stored and time to analysis.

Line 149: Could delete the word "retrieve"

Line 158: A chip was further "ground" not "grounded"

Line 161: was modified "by homogenizing bead tubes" delete "as follow" and delete "were applied"

Line 176: "were performed"

Line 182: "barcode" removal

Line 184: processed

Line 188: "further report specific abundances for each sample"
* * *
Line 135: To ensure or at least minimize contamination in the field, the hammer was thoroughly washed with surrounding sea water at each sampling point. The surfaces of all samples were then thoroughly brushed with sterile implements in the laboratory to eliminate all surface epiliths (and contaminants).

Line 137: Each sample was broken down to three pieces that were stored differently depending on downstream planned analyses. For each sampling location the samples used for mineralogical and biological analyses constitute biological replicates, we will clarify this point in the revised manuscript.

Air drying and alcohol preservation were both done in the field. Samples were transported in the dark at room temperature for 5 days before

Sea water samples were collected in sampling site K (west coast). Seawater was collected in a sterile polypropylene bottle, filtered on site on 0.22 μm sterile filter and stored at 4°C in the dark. After 5 days of transit at room temperature in the dark it was stored back at 4°C in the dark for an additional week before being processed.

Line 149-158-161-176 We will perform these changes

Line 182 There is one barcode per sample.

Line 184 -188 Of course we will correct this two, thank you.

Line 189: Suggest "Because this study focused on the most abundant OTUs and how they vary, rather than the rare biosphere, we filtered….to remove the few? rare OTUs"

Question how many rare OTUs were present?

Line 207: Suggest, "For comparison, raw sequences….

Line 212: delete "they were" substitute "and also" delete "as well"

Line 222: "ran"

Line 254-256: worded a little awkwardly,

Line 290: communities "in this study" are much more complex than the majority of literature "to date"

Line 294: proven that "some axenic" cyanobacteria delete "alone"

Line 296: other metabolic activities (of other co-occurring microorganisms) particularly those that result in pH changes….

Line 189 We removed a lot of rare OTUs, as mentioned in the text we analyzed only 11% of the total number of OTUs that were originally generated. However, the 89% OTUs that we removed accounted for less than 10% of the total sequences altogether (this information is line 193) .

Line 207-212-222 We will perform these changes

Line 254-256 We will reword this section as follow *"Isla de Mona was never continuously inhabited. The island was mostly used as a guard post for the Mona Passage throughout the 20th century, and declared a Nature Preserve in 1993 (National Parks Register, USA )."*

Line 294-296 We will perform these changes

Comment Lines 307-320 Nice discussion

Line 349: delete "does" ie., "one that not only represents" an initial set of pioneers...

Line 356: delete "always" substitute (has "so often" been described...)

And delete "who can" substitute "accounting" ie., "community accounting for"

Line 360. "were" instead of "was"

Question: are attempts in progress to isolate *Plectonema terebrans*? And are isolates available from other sources?

Minor editing lines 385 to 403

Line 404 delete "It becomes clear that substrate preferences" suggest substitute "Results suggest substrate preferences are found..."

Some rewording of section line 410 to 414, probably split into two sentences

Line 417 suggest "although the paucity of samples restricted our statistical power, we were still able to identify..."

Line 422: suggest delete "promiscuous" which is vague substitute "widespread across different substrate types"
* * *
Line 307-320 Thanks

Line 349 We will delete "does"

Line 356 As far as we know *P. terebrans* has always been described as an important player of endolithic communities so we will keep the wording.
We will rephrase the next sentence using "accounting for"

Despite our efforts we could never isolate *Plectonema terebrans*. We don't know of any other groups trying to do so at the moment, there are no isolate available.

Line 404-422 We will perform these changes

Comment: At the beginning of the manuscript "substrate preference" refers to clear numerical/ statistical occurrence (of particular endolithic cyanobacterial species relative to other species) in different mineral/ rock types. However at this point in the manuscript the term is "preference" is seeming to take on a more determinative meaning that is not yet demonstrated. Preference can be a tricky term to use. implying a more "decision based"

A particular species may seem to show a preference for a Mg or Ca cation containing mineral substrate based on occurrence/ density, but that does not necessarily imply a "preference" for Mg or Ca cations. That would need to be tested independently, as would the "preference" for anions.

The authors seem to understand this, but still sometimes fall into an overly interpretive phrasing implying metabolic/behavior from a distribution.

Certainly the data suggest some trends worth rigorously testing, (as has been done for *M. testarum* BC008) and it is critical to begin the determination of substrate preferences, by detailed investigations of naturally occurring distributions as the authors have done.

Line 447: in light of previous discussion would suggest changing "preference" to

"NR_OTU193) did show a higher rate of occurrence in calcium carbonates as compared to magnesium carbonate.

A careful re-reading and edits with particular emphasis on the implications and possible over interpretation of the term "preference" would be very beneficial.

Line 452: delete "we must assume" again data "suggests" but does not "prove"

Comment: Authors conclude that more factors may be involved in substrate preference that cation preference alone. A bit more discussion of what those other factors may be would be helpful.

Perhaps also in the introduction a short synopsis on thought / previous research as to "why" boring behavior is prevalent in some groups would be informative (ie., is behavior thought to provide protection from wave energy/ nutrients/cations/ light modification)?

We agree with referee #2 that we overused the word "preference" and we will carefully review the whole manuscript with that in mind, replacing the term by more accurate descriptive terms such as "rate of occurrence" or 'relative representation" each time that we can.

Following referee#2 advice, here is sentence we would like to add to the discussion to suggest alternative boring mechanisms:

"These contrasted findings highlight that there must be factors other than the cationic part of the mineral determining the excavation ability of a particular strain and that the boring mechanism proposed for *M. testarum* strain BC008 might be only incompletely described. *Other mechanisms have been suggested to*

*explain boring mechanisms which have been invalidated for the model organism M. testarum strain but may prove themselves valuable for other taxa. The dissolution of carbonate mineral by acid excretion was proposed by* (Haigler, 1969) *and* (Golubic et al., 1984). *This mechanism could involve spatial or temporal separation of photosynthesis vs. respiration by cyanobacteria or acid production as a byproduct of other heterotrophic bacteria activity* (Garcia-Pichel, 2006). *These hypotheses will need to be re-evaluated for other euendolith as well as in natural communities."*

Regarding the question as to "why" some groups of organisms do bore we will now refer to (Cockell and Herrera, 2008) who reviewed the question nicely.

Reference cited in the answer to referee #2

Cockell, C. S. and Herrera, A.: Why are some microorganisms boring?, Trends Microbiol., 16(3), 101–106, 2008.
Garcia-Pichel, F.: Plausible mechanisms for the boring on carbonates by microbial phototrophs, Sediment. Geol., 185(3–4), 205–213, 2006.
Golubic, S., Campbell, S. E., Drobne, K., Cameron, B., Balsam, W. L., Cimerman, F. and Dubois, L.: Microbial endoliths: a benthic overprint in the sedimentary record, and a paleobathymetric cross-reference with Foraminifera, J. Paleontol., 58(2), 351–361, 1984.
Haigler, S. A.: Boring mechanism of Polydora websteri inhabiting Crassostrea virginica, Am. Zool., 9(3), 821–828, 1969.

---

## Author Response (AR1)

Dear Steven,

We agree that both reviewers provided constructive suggestions, we are therefore happy to submit a revised version of our manuscript complying to their suggestions and based on the replies we previously posted.

We notably revised the manuscript by making a more careful use of the term "preference", providing a more detailed methods section, adding or rephrasing all ambiguous lines of discussion and fixed all the minor edits that were suggested to improve wording clarity.

We hope that you will now find our manuscript suitable for publication in your journal.

Sincerely,

Estelle Couradeau

[revised manuscript text omitted]

---

## Author Response (AR2)

Dear Steven,

We carefully reviewed our manuscript and performed an in-depth grammar and spelling check.

Sincerely,

Estelle Couradeau